# On the Understanding of the Unity of Organic and Inorganic Nature in Terms of Hegelian Dialectics

Cihan Cinemre 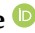

Department of Sociology, Faculty of Science and Letters, Mimar Sinan Fine Arts University, Istanbul 34380, Turkey; cihan.cinemre@msgsu.edu.tr

**Abstract:** The understanding of nature and its motion through Hegelian dialectics brings the notion of the organism that is intertwined with its inorganic nature. This notion is crucial first and foremost to comprehend life in its wholeness, as becoming that is in constant movement. To attain this comprehension, it is necessary to treat beings as entities invariably determining each other in their reciprocal relatedness. In this way, it becomes possible to set both the organism and its environment free of their fixity and quiescence. Within the work, to derive this mode of reasoning, the sciences and the dialectics are asserted in their unity. The relationship between the organism and its inorganic nature is one of tension. The organism in its finitude is in opposition to its inorganic nature; it is compelled to act to sublate the latter's independence, indifference, and exteriority for its self-preservation. This is called the melting of the non-organic into fluidity that renders the organism infinite. The relationship, as tension, elicits the notion of freedom; it signifies that freedom is not merely a matter of free will, it rather pertains to the organism's penetration into its exteriority, in which it can determine ever-changing goals for itself.

**Keywords:** dialectics; organic nature; inorganic nature; Hegel; freedom; assimilation





## 1. Introduction

In this work, I attempt to expound on the unity of inorganic nature with organic nature through the speculative philosophy of Hegel and its internal dialectics. This matters when attempting to examine nature in its wholeness. The notion of the whole is necessary to attain both the concept of the self and of the exterior that is constituted by that selfhood. A subject matter, if is not to be understood abstractly in its isolation, must be comprehended as a whole that constitutes nature both as a whole and as substance. The dialectical reasoning of Hegel lets us postulate the beings that move within this substance as interacting entities that determine each other through their reciprocal relationships, and in this way, we can obtain the notion of the whole that is in constant movement, as it truly is. This mode of reasoning is crucial to the constitution of the true ontology of any organism in its concreteness; in other words, to understand it as life, in its relatedness with its outer nature.

To construct that ontology, throughout the Section 2 of this work I aim to emphasise that the organism as such should be understood as a whole that is both distinct from and beyond the sum of the organs that constitute it. These organs mediate and regulate the external activities that render the organism a living being. This insight provides us with the subject's diremption into its organic and inorganic nature, which leads us to the notion of the organism as the unity of the interior and exterior. To attain the notion of the free organism, this unity also has to be understood as the unity of freedom and necessity. Accordingly, the sublation by Hegel of the observing reason that interprets freedom merely as the correlative of free will, as an assertion of contingency independent from any objective law, is examined throughout the Section 3 of this work.

The external activity of the organism is the fundamental element that brings the unity of freedom and necessity forward and the Hegelian notion of this activity is assimilation.

This notion, which is dealt with in the Section 4 of this work, particularly matters in renouncing any essentialist and idealist approach to the notion of the human and the relationship between the human and nature. Consequently, in this work, the animal is postulated in its unity and inseparable relationship with its environment and, through the dialectics, the ontology of the animal is put forward not in terms of presuppositions pertinent to its quiescent essence, but through its feeling of lack and its activity to satiate that lack.

Accentuation of the unity between organic and inorganic nature in this manner signifies the reciprocal relationship between the organism and its environment and indicates that they can only be separated at the level of abstraction; in reality, any organism is intertwined with its surroundings. Through this understanding, it becomes possible to move beyond both the view that considers the organism as being internally determined by its genes and the one that considers it as being determined merely by its experience of adaptation to its surroundings: any organism is determined by the interaction between its germination and its adaptation to the environment. Another way of attaining unity between the organism and its environment pertains to the organic being's emergence out of the inorganic through chemical processes. The unity between the organism and its environment, through their origins, actuality, and reciprocity, is dealt with in the Section 5 of this work.

## 2. Organism, Its Parts, and the Whole

For Hegel, "Nature is the Idea in the guise of externality" [1] (p. 418), "Nature is Spirit estranged from itself…" [1] (p. 14). What is implied by these phrases is neither the conceptualisation of nature by Hegel as a representation nor a dismissal of its objective being at the expense of an anthropocentric subjectivity. As William Maker says, Hegel neither denies the genuine existence of an independently given nature nor views the given nature as the product of thought. Although the notion of nature is the idea in the form of otherness, this does not imply that nature must be thought of as an idea or like an idea. In the *Philosophy of Nature*, initially, the content of the concept of nature immanently and explicitly is determined as not being thought or thought-like, and this corresponds to thinking of the Idea in the form of otherness. Hegel thematises nature as givenness and recognises it as otherness in an immanent but still nonreductive fashion [2] (pp. 4–10).

Here, the issue is rather to postulate nature in its unity with animals, through the notion of the organism in particular. Hegel's reasoning of nature is not anthropocentric, because he does not treat nature as the outer expression of the human mind. After all, Hegel's reasoning on nature, rather than consisting of an opposition between human and exterior nature, assumes an opposition between the organism in general and its exterior nature, which consequently culminates in their unity, and only through this unity does an animal achieve subjectivity. Hegel's endeavour is to constitute the notion of nature as a whole that is in constant movement and transition by means of the reciprocal relations between the beings that are active in nature. This endeavour is interpreted by Kirill Chepurin in this way: "…Hegel's philosophy of nature has for its subject not nature 'as such,' but rather a new, 'spiritual' nature…; the narrative of the identity of these two natures is not something given, but something constructed by spirit itself, retroactively" [3] (p. 302). This notion of Nature has implications for the notion of the organism; the latter is understood through the relationships that it forms with the outer through its activities—its simple moments. According to Hegel, "these simple moments are pervasive fluid properties, they do not have in the organic thing such a separate real expression as what is called an individual system of the shape" [4] (p. 166). Here, what Hegel indicates is that the organs of an animal have no independent beings of their own, and that they should be regarded as corresponding to the modes of motion or activity of the organism in its wholeness. In addition, the Hegelian notion of the organism does not consist in the organism's being a system of shape; it rather pertains to life and the fluidity and motion inherent to it:

> In the systems of *shape* as such, the organism is apprehended from the abstract aspect of a dead existence; its moments so taken pertain to anatomy and the

corpse, not to cognition and the living organism. In such parts, the moments have really ceased *to be*, for they cease to be processes. Since the *being* of the organism is essentially a universality or a reflection-into-self, the *being* of its totality, like its moments, cannot consist in an anatomical system; on the contrary, the actual expression of the whole, and the externalization of its moments, are really found only as a movement which runs its course through the various parts of the structure, a movement in which what is forcibly detached and fixed as an individual system essentially displays itself as a fluid moment. Consequently, that actual existence as it is found by anatomy must not be reckoned as its real being, but only that existence taken as a process, in which alone even the anatomical parts have a meaning [4] (p. 166).

Hegel deals with the relationship of parts to the whole and the difference between anatomy and organic life, which are essential to his philosophy of nature in general in the *Encyclopaedia* [5] (p. 204). When taken as a system of shape, as a combination of organs, the organism is postulated as a corpse. What Jean Hyppolite has to say about the subject matter is significant. He says that anatomy considers only cadavers, not living beings. When we cease to look upon the organs as the parts of a whole, we strip them of their specific beings, their organic beings. Organic being does not present distinct aspects that correspond to each other. Each part of the organism, Hyppolite says, is caught up in the movement of its resolution. The whole of the organic being is the movement and transition from one determination to another; it is already a concept, not a thing [6] (p. 253). Then, there is a twofold aspect of living organisms, as Allegra De Laurentiis puts it: one is physicality and the other is immateriality [7] (p. 23). In Alison Stone's words, if the idea is to realise itself, it must not only assume the shape of the organism but must advance to what Hegel calls cognition [8] (p. 141). Thus, when the animal is postulated in its externality merely as the sum of its parts, as a physical being is not grasped as the correlative of life, or cognition but is reduced to the sum of its systems, as if it were a machine.

One should not understand the organism as natural or physical: it is universal or reflection-into-self, which means that the organism is the notion of genus and is in motion, and cannot be reduced to an anatomical system. The notion of the organism should be understood in terms of the whole in motion, both internally and externally. The moments of the organism manifest themselves as external within the parts of the whole organism and as a movement among them; in this way, an organ or a system of organs appears as a moment of fluidity in its wholeness. Consequently, anatomy is not the real existence; the real existence is the life in motion, as an actuality corresponding to its notion. The subject constitutes itself by sublating its own anatomical entity, and its constitution as the Spirit is defined by Gilles Marmasse as the activity that brings the natural externality back to the living whole. The Spirit is the sublation of naturalness as pre-supposed given manifold. The Spirit is neither inert nor an activity without a subject; it is a subject in action that constitutes itself theoretically and practically as the unifying principle of naturalness—both the exterior nature and itself as a simple presupposition. Otherness is sublated by the subject; therefore, the latter is actualised. Through sublation, the subject renders the object as its ideal, making it the subordinated material to affirm its own fulfilment. This is the negation of negation through which the subject takes charge of the given manifold and constitutes itself as a totality. The subject negatively suppresses the opposition between itself and its other in the act of sublation and, at the same time, positively, by subordinating the other, establishes the unity between itself and its other [9] (pp. 19–20).

Hegel emphasises that a law of being should treat the organism in its wholeness. He says that, "in such a law they [moments] are asserted of an outer existence, are distinguished from one another, and neither aspect could be equally named in place of the other". Therefore, when the moments of the organic inner are taken in their isolation and as fixed, they cannot bring in the elements of the law. For an organic being is in itself the universal, its essential nature consists in its moments being universal in actual existence, "in their being pervasive processes, but not in giving an image of the universal in an isolated thing" [4] (pp. 166–167).

In the *Encyclopaedia,* Hegel puts this straightforwardly: " … [T]he various parts and members of the organic body have their subsistence only in their union, and cease to exist as such if they are separated from one another" [5] (p. 196). The Hegelian outlook on nature and the notion of the unity of organic nature and inorganic nature internal to this essentially revolve around the notion of the whole, as will be emphasised throughout this work. Just as Errol E. Harris suggests, Hegel constantly seeks to establish the primacy of the whole. His insistent reference to the concept stresses the whole [10] (p. 196). This means that these moments, in their externality, have to manifest themselves as elements of life as a whole; in other words, they should not be thought of as working separately throughout the activity of the organism as life. Cinzia Ferrini explains how Hegel understands the relationship between parts and the whole:

> …Hegel's point is that in truth the parts are mutually related as interdependent *moments* of one whole. Hence their real differentiation and division is *ideally* and *necessarily* reintegrated into the unity of their common purpose, namely, the conservation of the organism in a state of functional activity, directed so as to cause feedback from the outside world [11] (p. 204).

Life dialectically is the whole, both in the sense that it signifies existence beyond the sum of the parts that it embodies and that the notion of life does not merely connote the individual as a whole. Wholeness also entails the whole as the genus, which comes into being through the mode of action of its individual units in their environment. In this way, the notion of life as unity between an organism and inorganic nature is attained. As stated by Michelini, Wunsch, and Stederoth, according to Hegel, the idea of the living being must be understood as a wholeness that organises its components as members. For Hegel, the organism is, on the one hand, the self-organising and self-preserving being and, on the other hand, relates to the other while always remaining itself. It withstands alterity in itself, copes with the contradiction in itself [12] (p. 6). Thomas Posch agrees with this view when he says that, according to Hegel, all entities exist exclusively by virtue of their relatedness to other entities [13] (p. 191).

To comprehend the notion of the organism as Hegel understood it, the *Philosophy of Nature* provides us with a definition in terms of its relationship with inorganic nature. Since the animal, as life, is an immediate being, it is discrete, finite, and particular:

> Life (*Lebendigkeit*), tied to the infinitely many particularizations of inorganic and vegetable Nature, exists always as a limited species; and these limitations the living creature cannot overcome…Life, which receives these powers of Nature (*Naturepotenzen*) into itself, is capable of the most diverse modifications of its structure (*Bildung*); it can adapt itself to every condition and still pulsate among them, although the universal powers of Nature (*Naturmächte*) always retain their complete mastery [1] (p. 417).

Immediate existence corresponds to the animal's bodily, physical, and natural existence. An organism's individual immediate existence arises from its connection with its inorganic nature. Then, the freedom, discreteness, particularity, self-preservation, and animal's constitution as an organic being are possible only through the inorganic nature: as a genus, the animal is always limited. This is a twofold limitedness: first, the animal's physical capacity limits the possible relationships that it can form with nature and, second, its inorganic nature limits the animal's natural life by rendering it finite through elemental forces, disease, and death. Since the organism acquires its power from its inorganic nature, an element of reflectedness into self emerges. Self-reflectedness is also motion as the life and wholeness of the organism. Hegel puts this motion forward as the infinite process in which individuality determines itself as particularity and finitude; then, it negates this to re-establish itself at the end of the process as its beginning. As a self-related negative unity, it becomes subjective [1] (p. 273). Inorganic nature creates a sphere for the animal that can constitute itself in manifold ways: it sets animals free, forming the unity of necessity and freedom. This freedom exists in unity with its opposition because, as the organism gains

the capacity to adapt itself to diverse conditions, it simultaneously acquires the capacity to incorporate the forces of nature into itself. Thus, although tension defines the relationship between the animal and its environment, the former still pulsates among the conditions of the latter; still, the organism is in a subjected state against the universal forces of inorganic nature.

Hegel asserts that life, as a process, is its own conclusion with itself, and one of the processes that occurs within the living being is the latter's diremption and the turning of its own corporeality into its object or into its inorganic nature [5] (p. 292). Catherine Malabou agrees with this view when she says that the individual considers himself as a being divided into an exteriority, that is his body, and an interiority, his inner being [14] (p. 66). Hegel's reflection on this specific process is significant in the sense that it consists of essential insights into the content of the unity between inorganic and organic nature: with corporeality being inorganic, as exterior to the spirit. Then, the distinction between life and corporeality is apparent; in their relationship, the living being is separated from its anatomy, which is now objectified as the former's inorganic nature, but they are nevertheless united, to be understood as life. The diremption of the living being brings forth a contradiction that is internal to the relationship between the being's corporeality and its immaterial existence. De Laurentiis refers to this contradiction as the notion of mechanical soulfulness. The notion signifies the spirit's rootedness in inorganic nature, even in its mechanical subsystems. Here, de Laurentiis says, Hegel draws attention to a fundamental asymmetry present in the soulful dynamics of natural mechanisms. This is the asymmetry that anticipates the imbalance characteristic of living bodies: as long as the mechanical system exists, the cohesion of its parts is the dominant force, and its negation is reactive. For the mechanical system to endure, it must be continuously negated [6] (p. 138).

Owing to this insight, it also becomes possible to understand the relationship between parts and the whole in terms of its implications for the notion of the unity between organic and inorganic nature. As Hegel says, inorganic nature, in its relative externality, enters into the distinction and antithesis of its moments, but the activities of these separate moments or members are, in fact, the activities of a single, discrete subject [5] (p. 292). Then, these moments or members are inorganic in the sense that they do not possess the capacity to live independently; they are both distinct from and united with the living being. In the words of Jane Dryden, this is the selfhood that "allows us to acknowledge otherness within us while still having enough unity for agency" [15] (p. 1). This insight does not merely pertain to a judgement drawn up by way of speculative philosophy. Dryden emphasises the growth in our understanding regarding the biological relatedness that directs us towards an ontology of the self that is constituted by its organic and inorganic others; however, this self possesses the coherence of the phenomenologically unified subject [15] (p. 6). Regarding this subject matter, what Dryden has to say is intriguing:

> Our gut serves for us as a kind of ambiguous other, one which is sometimes experienced with hostility. Rather than endorsing that hostility, Hegel, the philosopher who calls for the unity of unity and difference, is an ally in giving us a theoretical language that helps us to be at home with our gut  [15] (p. 19).

It is necessary to separate the organic from the inorganic at the level of abstraction and deal with them in their distinction and opposition to grasp the significance of their unity. To grasp the extremes in their distinction, they must be understood through their predicates as determining the organic being as discrete and the inorganic nature as continuous. Daniel Lindquist indicates this mode of reasoning as being internal to Hegel's philosophy: "What Hegel means by the 'continuity' of the parts is that they all are 'the same thing' in the sense of all falling under a common concept, which common concept enables them to be 'quantitatively distinguished' or counted" [16] (p. 389). In sum, since any organ is not a living being and is not discrete within the whole, the organic contains the inorganic. Therefore, inorganic nature is both external to the animal and is an internal part of the organicity of the animal.

### 3. The Unity of Necessity and Freedom

We must recognise the unity between organic nature and inorganic nature as the correlative of a more general unity between freedom and necessity. This recognition is essential to obtaining a true understanding of subjectivity and the manner by which it constitutes life in its relationship with the outer. Hegel expounds the unity of necessity and freedom most clearly in *Encyclopaedia.* Here, the crucial issue is the assertion of the contingent as possessing the ground of its own being not within, but elsewhere. In the sphere of the practical, the issue should be going beyond the contingency of the will and the freedom of choice. When one speaks of the freedom of will, people think of it as the correlative of the freedom of choice, which is will in the form of contingency. However, this is not freedom; it is freedom in the formal sense, which proves to be a contradiction since the content and the form are still in opposition. The freedom of choice is given and grounded in external circumstances. Freedom, in its relationship with such content, consists only in the form of choosing, which is merely a formal freedom [5] (pp. 218–219). The way in which Hegel understands nature matters to grasp what freedom really is, in its concreteness. When the notion of freedom is enunciated as being in unity with necessity, this does not merely signify that freedom is conditioned by necessity; instead, this unity indicates that necessity, as the exterior of the organism, determines freedom. Freedom depends on the necessity to develop, be actualised, and become real.

A notion through which the unity of freedom and necessity is substantiated is reasonable will. This notion refers to the dialectical law that if the will, as the correlate of freedom, is actualised, it should be rational. When Gerad Gentry says "the structure of reason is a dialectic of free lawful purposiveness" [17] (p. 167), this simply signifies reasonable will. This is to say that the will must be aware of the laws of objectivity that are necessary for its realisation and is immersed in the same laws. Christopher Caudwell speaks of the realisation of freedom as an advance in the consciousness of necessity. According to Caudwell, the development of life is determined by the tendencies of life. The development of life produces an increasing synthesis between the environment and life, called the consciousness of necessity. This development not only secures the transformation of the tendencies, the alternation and the elaboration of goals, but it also ensures the congruence of changes to goals. Life's ability to realise its End invariably increases [18] (pp. 171–172).

To comprehend the unity between organic and inorganic nature, it is not adequate to consider only the inner being of the organism, since it has an outer being as well. Hegel says that the outer, when considered in general, is the structured shape:

> . . . [The] system of life articulating itself in the *element* of being, and at the same time essentially the being *for an other* of the organism—objective being in its *being-for-self*. This *other* appears, in the first instance, as its outer inorganic nature. . . [the organism] is at the same time absolutely for itself, and has a universal and free relation to inorganic nature [4] (p. 170).

The outer, as a structured shape, both implies the determinateness of the objective world and the determinateness of the self-externalisation and self-objectification of the organism and of its concrete being. Additionally, *the articulation of the system of life in the element of being* corresponds to the organism putting itself forward in the inorganic world—in its other as substance, which is the organism's being for an other. Hegel mentions that the organic, in relation to its inorganic nature, is universal and free; this implies the subject's self-objectification within its interaction with inorganic nature. Here, the term free implies the organism's freedom to separate itself from its external nature, to determine a specific End for itself, both external and internal, to determine the means and the method to achieve that End, and the self-reflectedness with which it changes both its inner and outer nature.

What is preserved in the unity of organism and the inorganic nature is the self, the notion that brings up the issue of unity of necessity and contingency. Since self-preservation is initially an individual activity, at first, any relationship with necessity is not contained in this. Therefore, Hegel says that the nature of being is to conceal this necessity and present

it in the form of a contingent relationship. This occurs as a result of the detachment of the self-preservation of the organism as an individual from the preservation of life in general. The individual will is considered in its indifference as being capable of freedom, as if its relatedness precludes individual free will: "Thus it presents itself as something whose Notion falls outside its being" [4] (p. 158). Since the reason in question here is instinctive, the laws that rule the relationship between the organism and objectivity are regarded as antithetical to freedom. Then, it is impossible for instinctual reason to reach the Notion of the objective world that the organism is immersed within; this is why necessity is negated. Hegel remarks that reason, as instinct, remains on the level of a state of indifference and mere being; therefore, the thing and its notion seem to be mutually exclusive. This is the observing reason's view of the organism: the organism remains outside of its own Notion; the actions of the organism seem to be actions of an arbitrary unrestrained will and cannot be included in any notion. Observing reason makes a distinction between the Notion of End and being-for-self and self-preservation and this is a non-existent distinction: "...[T]he said act and the End, falls asunder for the consciousness" [4] (p. 158). Hegel expresses this lawlessness even more specifically: since the self-preservation of the organism as an individual or genus is directed towards an immediate expression of a contingent necessity, it remains unrestrained by any law; this is also due to the exclusion of the universal and Notion. Then, the activity of the organism lacks a content of its own and the unity of being and notion is precluded. However, this relationship with an other is an activity, and the being-for-self is not merely specific but universal, and its End is not external [4] (pp. 158–159).

One must sublate the way that observing reason views the world to attain a true notion of the latter. Because for observing reason, the organism has the appearance of a relationship between two fixed moments as immediate beings. According to this consciousness, Notion is the inner and the actual is the outer [4] (pp. 159–160). As Andrea Gambarotto says, the laws that observing reason draws up are inadequate to account for the living organism, which is the concrete manifestation of what Hegel calls the concept. This is a form of internal unity that manifests itself through its relationship with an other. For Hegel, observing reason is limited, because it cannot be elevated to the level of concept to provide a truly holistic account of living beings. Therefore, observing reason assumes that vital properties are isolated elements and their concrete relations with the whole, of which they are a part, are ignored [19] (p. 118). Therefore, Hegel goes beyond observing reason.

The first moment of the sublation is to approach the organism through its simple essentialities that are asserted by the law as the relationship between the organism and its inorganic nature. For Hegel, beyond observing consciousness, the internal and external do not appear as self-subsistent things and the universal does not exist outside these extremes: "On the contrary, the organic being in its absolute undividedness is made for the foundation, as the content of inner and outer, and is the same for both." For Hegel, inner substance is the unitary soul, the pure Notion of End, or the universal. The outer, in contrast with the inner, subsists in the quiescent being of the organism. The law of the relationship of the inner with the outer asserts its contents as universal moments and simple essentialities. These initial simple properties are sensibility, irritability, and reproduction. These signify the animal organism, not the organism in general. Vegetable organisms express only the simple notion of the organism, which does not develop these moments [4] (pp. 160–161). Then, through its organic properties, we begin to conceive the organism as discernible from its other, as the sphere in which its absolute freedom will find expression. In Hegel's words, the extreme of being-for-self is the pure singular, the simple negativity; that is to say, it stands opposed to inorganic nature as its other and is absolutely free, owing to the inorganic nature that it is indifferent towards and secured against [4] (pp. 170–171). Hegel says that the freedom of the organism is also the freedom of its moments (sensibility, irritability, and reproduction), in their appearance as outer existence and being apprehended as such. In their activity, these simple moments relate to each other to acquire wholeness. This is the movement by which absolute freedom appears:

> This Notion, or pure freedom, is one and the same life, no matter how many and varied its shapes or its being-for-another; it is a matter of indifference to this stream of life what kind of mills it drives [4] (p. 171).

Here, the organism, as life, is indifferent towards the particularities of the sphere that it acts upon. Organisms, in the face of their environments, are free, but not free enough. As Lindquist says, no living being lives because its environment forces it to do so. When an animal acts on an opportunity, it creates an environmental cause to produce an effect. Therefore, the living being is free in a way that inorganic nature is not. It lives since it is not pushed around in space by the force of blind necessity. However, this freedom is limited in the sense that a living being lives as its environment allows; even living beings are determined by the kind of environment that they inhabit [16] (pp. 390–391).

The living being, when determined by its environment, indicates the sublation of the abstract freedom of the "I." This sublation is necessitated by the law of the unity of necessity and freedom, which finds its specific expression in the relationship between the animal and its inorganic nature as the notion of evolution. This notion presents Life as the history of the more intense penetration of the animal into its environment, and this history involves the animal's growing capacity to act freely in conjunction with its increasing awareness of its surroundings. Then, Life signifies an animal's increasing ability to carry out its ever-changing goals as it diversifies its actions. The process, known as evolution, signifies the perpetual conflict between the organism and its environment, a conflict which is constantly sublated as the organism expands its field of action and advances its species properties. Therefore, in both its unrest and its quiescence, the organism is inseparably tied to its inorganic nature as its necessity. Hegel puts this reasoning forward as the unity between willing and outer reality, a crucial aspect of which is the experience of negativity as need:

> People believe that it is in the will that they are free, but it is just in willing that they are in a relationship with a reality outside them. It is only in the reasonable will, which is theoretical, as in the theoretical process of the senses, that man is free. What is primary, therefore, in animal appetite is the subject's feeling of dependence, that it is not for itself, but stands in need of an other which is its negative, and this not contingently but necessarily; this is the unpleasant feeling of need [1] (p. 387).

Freedom is the essential notion that defines any organism. It is an organism's ability to constitute itself as being discernible from its environment, both as an individual and as a genus—as a matter of fact, the necessary practical distinction between the two is an essential condition of an organism's freedom. Out of freedom there emerges what Hans Jonas calls selfhood: The "profound singleness and heterogeneousness within a universe of homogeneously interrelated existence" [20] (p. 83). Evan Thompson reflects on the free organism through the notion of autopoiesis: the living organism stands out from a given chemical background as a closed network of self-producing processes that actively regulate its encounters with its environment. However, this self-isolation does not mean the organism is independent of the world. The organism is in the world and of the world, and its identity has to be enacted in the process of living. Autonomy, rather than the organism being exempt from the causes and conditions of the world, is an achievement that is dependent on these causes and conditions [21] (pp. 149–150). Since the organism depends on material exchanges with its environment and on the metabolic relations that it forms with it, this is needful freedom, as Jonas states: the organic form has a dialectical relation with matter [20] (p. 80).

This crucial view implies that the will does not involve freedom; it is only possible to speak of willing as freedom when it is directed outwards and is related to reality. Here, the theoretical process of the senses manifests itself as the middle term of the inner and the outer and, through the reasoning of the whole that is perceived by that sensuousness, the

reasonable will becomes a will that recognises necessity, a will that immerses itself into the substance that it moves within; then, it becomes a true will.

## 4. Assimilation

The dialectical understanding of animals signifies them as possessing a self-feeling that is discrete, particular, and self-subsistent. This is how an animal is recognised as a distinct being. On this subject, Hegel says: "Because the animal is a true, self-subsistent self which has attained to individuality, it excludes and separates itself from the universal substance of the earth which is for it an outer existence" [1] (p. 355). Then, Hegel indicates the organism's relationship with its environment as a means of its formation as an individual. The world exterior to the organism is not dominated by it; it is the negative of the animal and, in this way, the non-organic nature of the animal is individualised through their relationship, since the latter is not separated from the Element. According to Hegel, the notion of animal is this relationship between the organism with non-organic nature. The animal is the individual subject that enters into a relationship with individual objects; thus, the organism is differentiated from the plant, which only enters into relationships with Elements [1] (p. 355).

Here, what is in question is the actual organism. The essential aspect of this organism is its being-for-self, which also connotes that it is immersed in the objective world. This is why Hegel emphasises that the actual organism is the middle term that unites the being-for-self of life with the outer in general, and with being-in-itself [4] (p.170). Being-for-self is identified with activity, freedom, and universality. For Hegel, being-for-self is "the inner as an infinite One." It takes the moments of the shape from their connections with outer Nature and their subsistence into itself [4] (pp. 170–171). The notion of infinite One is as follows: it is the I that is we and we that is I [4] (p. 110), and it moves nature out of its independent existence. This is the infinite that Findlay speaks of as "the deepest essence of our conscious personal being: life in the more definite medium of what exists out there is a monogram, an analogue, of that more ultimate indefiniteness that we experience as ourselves" [22] (p. 92). Now, the infinite as One is the subject that is to be understood through its relationships with an other, through the unity between outer and inner. As Hegel puts forward: "A being which is capable of containing and enduring its own contradiction is, a *subject*; this constitutes its infinitude" [4] (p. 385). Here, the contradiction is between this lack and overcoming it. In the words of Luca Illetterati, the organism, while overcoming its condition, passing the limit, and satisfying its restlessness, experiences a tension that compels it to engage the outer world; in this way, it becomes what it really is [23] (p. 197).

Here, the free will of the animal and its subjectivity come into play since the animal has the capacity to determine and construct its surroundings. In other words, to a certain extent, the environment is individualised and the middle term that mediates this relationship is stimulation; this indicates that any organism does not attain its specific mode of existence in a haphazard manner, but through stimulation by its specific nature. Hegel expounds this relationship:

> The animal can be stimulated only by *its* own non-organic nature, because for the animal, the opposite can be only *its* opposite; what is to be recognized is not the other as such, but each animal recognizes its own other, which is precisely an essential moment of the peculiar nature of each [1] (p. 390).

In other words, in the relationship of opposition, one does not only encounter an other, but its other [5] (p. 187). This means that, within the relationship between organic nature and inorganic nature as opposites in unity, the otherness is not general, but particular:

> The purpose of philosophy is…to banish indifference and to become cognizant of the necessity of things, so that the other is seen to confront *its* other. And so, for instance, inorganic nature must be considered not merely as something other than organic nature, but as its necessary other. The two are in essential relation to one another, and each of them is [what it is], only insofar as it excludes the other

from itself, and is related to it precisely by that exclusion. Or in the same way again, there is no nature without spirit, or spirit without nature [5] (p. 187).

Hegel further elaborates on the relationship that constitutes individuality. The process of individuality moves in a closed circle: this is the sphere of being-for-self of the organic being. This being-for-self is its Notion; then, its essence (*Wesen*), as its non-organic nature, is individualised for the organic being [1] (p. 355). Then, since the End of the organism is its own self, and to achieve this End it should perpetually withdraw from itself to move into inorganic nature and eventually return to itself, movement as self-reflectedness is what constitutes the organism as being-for-self.

Assimilation is the crucial notion by which to apprehend the organism constituting itself as individuality and being-for-self and the way in which these processes are understood by Hegel. He speaks of two distinct processes of assimilation. One is formal, through which instinct "impresses its specific nature (*Bestimmung*) on the details of its outer world and gives them, as material, an *outer* form appropriate to the end, living their objectivity untouched." The other is real assimilation, through which the instinct, "individualizes inorganic things or relates itself to those already individualized and assimilates them, consuming them, and destroying their specific qualities..." Breathing, thirst, and hunger are elements of real assimilation [1] (p. 390). Yrjö Haila and Richard Levins indicate the matter of assimilation through the openness of the organism and the relationship of exchange it has with its environment:

> ...[A] multicellular organism is a complex system whose survival and reproduction depends on the exchange of material and energy with its surroundings. Unlike non-living materials and artifacts which are preserved by isolation from the environment, organisms have to be open to the outside. [24] (p. 136).

This openness is what Hegel puts forward as the notion of life: it is the subject of these moments of totality and the development of the tension between itself and exteriority. Life is the perpetual conflict through which this externality is sublated [1] (p. 390). Here, the unity between inorganic and organic nature is manifested as the totality of life. This is the notion of life as the movement by which the organism and its environment constantly unite and fall apart. Ferrini speaks of this motion in this way: " ... [L]ife begins from an essential though abstract principle, distinguishes or particularizes its components, and then reintegrates these real divisions within the original essential principle to form a concrete living individual" [11] (p. 204). Here, tension manifests itself as the notion used to define the relationship between the animal and its environment. As Wes Furlotte says:

> As sentient, animality is not poured out in the plenum of material environment such that it is unable to distinguish itself from the manifold of objects in which it is immersed; rather, it carves out a negative unity that distances it from that context and into which those external determinations are drawn and experienced in sensations of pleasure, pain, etc. The uniqueness of sensibility and feeling is not such that the environment imprints itself on the animal, the case is rather the opposite. The animal assimilates the environment to itself, transforms the latter into an inner, qualitative affection of its own [25] (p. 60).

There is a distinction and opposition, which are constantly sublated, between animality and the inorganic nature. The environment stands as the opposite of the animal, but at the same time, it is constantly internalised. Then, the exterior is invariably construed as the essence of the animal.

The real assimilative process, according to Furlotte, is the real practicality of assimilation, and the digestion of externality: in that sense, every fibre of the animal is transformative, and the organism breaks down the externality to its own ends. The animal organism constantly seeks to overcome the otherness of its environment. This will of the animal characterises its life: it is the perpetual interface of an assimilation that is a point of strife and acute tensionality [25] (p. 67). When Hegel expands on the Notion of digestion, he propounds that the essential moment that it consists of is the process. Here, the organism

is in tension with its own inorganic nature; it negates the latter and makes it identical with itself within the immediate relationship between the organic with the inorganic. The former is the melting of the non-organic into organic fluidity. The grounds of every reciprocal relationship between these two are the absolute unity of substance and, through this, the non-organic becomes transparently ideal and non-objective for the organic being. Non-organic nature becomes corporeality, belonging to the subject through the alimentary process [1] (p. 397). Once more, tension appears within the relationship between organic and inorganic nature. However, this tension results in identification; the inorganic becomes contained in the fluidity of life.

Hegel then goes beyond the opposition between organism and the inorganic nature. The individual stands against inorganic nature, but the connection between the two is absolute, inseparable, internal, and essential. Because the organism possesses negativity within itself [1] (p. 381), the animal is inseparably attached to inorganic nature. This is the life-as-motion that comes into being in and through contradiction:

> Now since the organism is directed towards the outer world as well as being inwardly in a state of tension towards it, we have the contradiction of a relationship in which two independent terms appear mutually opposed while at the same time the outer must be sublated. The organism must therefore posit what is external as subjective, appropriate it, and identify it with itself; and this is *assimilation* [1] (p. 381).

The simultaneity of intertwining and the distinction between environment and organism is also conceptualised as the tension between them by Caudwell. There is a tension between life and the environment; this is not merely the incursion of life into a static world but is the development of contradictions in the matter that separate the living and non-living matter, which stand as opposite poles: as life against the environment and man against nature. However, these opposites interpenetrate and, through this, the increasing complexity of the world of nature is developed [18] (p.173). The determination of the relationship between organic nature and inorganic nature in terms of the notion of tension signifies life as becoming. When life is understood as becoming, then it becomes possible to understand that necessity brings forth greater freedom, and this further connotes the subject having a rational consciousness. What is in question is not abstract freedom inherent to the organism, but is concrete freedom, as actualised through the organism's external activity. The activity that perpetually reconstitutes the organism's essence provides it with a rational consciousness and sublates the finitude of the organism.

As Terry Pinkard says, animals deny the self-sufficiency of worldly things [26] (p. 19). However, neither worldly things nor the animal that denies their self-sufficiency are quiescent beings: they reciprocally determine each other. Nature is not merely receptive in the face of the actions of animals. As Engels says, nature takes her revenge after each human victory over nature. Each victory initially produces the desired effects of the human action, but then quite different and unforeseen effects manifest themselves. Humans by no means rule over nature like a conqueror; humans belong to nature and exist in its midst [27] (pp. 460–461). At every moment, nature, through its unruliness, poses new problems for humans that must be overcome.

## 5. Reciprocity and the Unity between the Organism and Its Environment

Hegel's speculative philosophy of nature that postulates the relationship between the inorganic and organic nature as pertaining to the notion of the whole and the reciprocal relationship between the organism and its environment is crucial when constituting a dialectical apprehension of biology. A lucid account of the unity of organism and environment in Hegelian dialectics is given by Harris:

> The inorganic substratum. . . must be seen as the universal body of life as such, the precursor of self-conscious mind. In this again,. . . he prophetically anticipates the scientific conclusions of our own day. . . [H]e describes the geological system

as pregnant and permeated with life, and he is constantly and acutely aware of the indissoluble continuity and interdependence of organism and environment…Everything is seen and described as interdependent with everything else. Earth, sea, and air interpenetrate and interchange…In short, the transition from inorganic to inorganic is continuous and the frontier between the two is blurred and shadowy [10] (pp. 199–200).

Here, the decisive issue is that the organism and its environment are separable only as an abstraction and, in the concreteness of real life, they are in unity. Richard Levins and Richard Lewontin [28] assert the necessity of making a distinction between the internal and external in the advances made by modern biology. However, this distinction is *bad biology* regarding the problems of today's scientific problems and presents a barrier to further scientific advances [28] (p. 31). Yuliya Yurchenko is in line with this insight when she says that humans and nature are separable only on the level of abstraction [29] (p. 38). Lewontin emphasises Darwin's revolutionary leap in understanding the relationship between the organism and its environment. Prior to Darwin, there was no clear demarcation between internal processes and external ones; there was no distinction between the living and the dead, animate and inanimate. Then, Darwin ruptured this intellectual tradition by alienating the inside from the outside by separating the internal processes from the environment. Without this alienation, Lewontin says "we would still be wallowing in the mire of obscurantist holism that merged the organic and the inorganic into an unanalyzable whole" [30] (pp. 42–48). However, this initial advance made by Darwin now bars further theoretical progress. Fred Magdoff and Chris Williams state the same problem in this way: "The near absence of holistic thinking in Western science—considering the whole system and examining the interactions occurring within it—continues to be responsible for the slow progress in many branches of science" [31] (p. 219).

According to Caudwell, the organism should not be understood as separated from its environment; these two should not be supposed as mutually exclusive opposites, as one is living and changeful and the other inert and changeless. Then, Caudwell goes on to say that, when these two are separated, "neither environment nor organism are real environment or real organism, for they are only *really* real as related parts of one real universe" [18] (p. 179). Therefore, it is not right to identify the organism with vitality and liveliness and the inorganic with quiescence and lifelessness. On this matter, Alexander Bogdanov states that the separation of psyche and matter by a whole chasm and the consideration of them as things that are absolutely heterogeneous and completely incommensurable are the habits of the old philosophy. In fact, there is no such absolute distinction. This interpretation removes the boundary that was set up by the old worldview between dead inorganic nature and living organic nature, viewing the former as absolutely lifeless and devoid of any organisation. The technical evidence negates this view: "…[L]ife is constantly maintained at the expense of the material of inorganic nature, and dead matter, being assimilated by organisms, actually turns out to be capable of life" [32] (p. 232). This shows us that one can grasp the organism and its environment only as the constituents of the whole, only as interacting real entities, not as isolated, independent, and abstract ones; they are real entities that exist in and through action and, when understood in this way, attain their ontological concreteness.

Life, enacted in the unity between organic and inorganic nature, comes into being through the reciprocity of heredity and adaptation and the external and internal. Life is not merely a process that is originated and proceeds within the interior of the animal. With respect to this issue, Caudwell asks: "if the quality is acquired can it be inherited?" Caudwell says that this question is meaningless since all characters are germinal responses to an acquired situation [18] (p. 180). Rob Wallace summarises Caudwell's point of view in this way: genes are abstractions and their actual effects are context-dependent. Genes' impact on the world can only be found in their interpenetration with other genes and other sources of inheritance found in the environment [33] (p. 31). The unintelligibility of animals merely in terms of their hereditary properties, and the only true ontology of

theirs being possible through their relatedness, are postulations that are articulated by B. Zavadovsky as well. He speaks of the need to stop conceiving of the biological process of development using over-simplified mechanistic attempts as if they were the result of the physical influences of external surroundings, or of similar physical and physicochemical processes inside the organism or its genes [34] (p. 76).

This matter is a fundamental aspect of the works of scholars who seek to dialectically understand nature and its internal motions. Posch maintains that modern biology has to admit that concepts such as energy, entropy, and genetic code do not exhaust the essence of living beings [35] (p. 66). Lewontin, in his *Triple Helix,* comprehensively criticises the view that prevails in modern developmental biology, which reduces the development of the organism merely to the activities of cell organelles, and genes, and elucidates organic nature and its coming into being merely in terms of an internal process [30]. Salvatore Engel-Di Mauro propounds that the development of materialist dialectics in the biophysical sciences should emphasise interconnectivity and transformations in the biophysical processes [36]. According to Haila and Levins, the behavioural repertoire available to any person will differ from person to person, because of their immediate surroundings at work, in their communities, and in their larger society [24] (p. 147). In *The Dialectical Biologist*, Lewontin and Levins maintain that the development of an individual organism cannot be solely the unfolding and unrolling of an internal program. The organism is the consequence of a historical process that lasts from birth to death, and, at every moment, genes, environment, chance, and the organism as a whole participate in this process. The environment and the organism actively codetermine each other. External and internal factors, genes, and the environment, through the medium of organisms, act upon each other [37] (p. 89). The reciprocity of the organism with its environment and idea of the organism being the subject constructing its own environment are also central themes of *Biology Under Influence*. Addy Pross asserts that life is more complicated than a representation provided by a string of 3 billion letters. The spectacular advances made by molecular biology, which is reductionist in its approach, have not opened the gates of the promised land [38] (pp. 114–115). Magdoff and Williams deal with this issue using the concept of biological determinism. The idea of genetically determined social traits and talents determining human beings and whether they are going to be successful, even before the moment they are conceived, has been already debunked as we more learn about genetics and the role of DNA [31] (p. 207). Thompson approaches the same matter using the concept of *genocentrism,* which holds that the gene is the fundamental unit of life and the primary unit of selection in evolution. Thompson says that the concept of DNA as an information store is an oversimplification that has little predictive or explanatory power and obscures the understanding of the dynamics of autopoiesis, reproduction, heredity, and development [21] (pp. 183–184). Without a doubt, one of the most compelling accounts of how the organism is physically determined by its environment, which also happens to be a human construction, was provided by Friedrich Engels in the *Condition of the Working-Class in England* [39] (pp. 295–583).

The inseparability of the animal from its environment and the unity of organic and inorganic nature, in general, is not merely about the reciprocal relationship between the organism and its environment or the organic being creating a world after its own image. The organic emerges from the inorganic nature by way of the chemical processes that contingently engender life. Hegel says that the chemical process, in general terms, is life. The underlying reason for this statement is twofold: through the chemical process, the individual body in its immediacy is both produced and destroyed; thereby this notion leaves the stage of inner necessity and is made manifest [1] (p. 269). This is the notion actualised, which is free of the constraints of the being in itself. Then, Hegel speaks of one of the fundamental principles to discern an ordinary chemical process from Life: in the chemical process, the beginning and the end of the process are separate and distinct, and this constitutes its finitude, which keeps it apart from Life [1] (p. 269). Therefore, the organism, being both the cause and effect of itself, is distinguished from any regular chemical process.

Another aspect of this distinction is the dependence of any chemical process on externality to commence; additionally, at the end of the chemical process, the products are mutually indifferent, which means that the process cannot live up to the Notion as one. As R. D. Winfield puts it, life also depends on external conditions. However, an organism sustains itself through constant internal transformations that renew its constituents and the process of transformation [40] (p. 390). Winfield states that life must enlist the chemical process because only the chemical process involves transformations that are not superficial. The proper fortuitous combination of the plurality of chemical processes may engender a process, the inputs and outputs of which are identical. Thereafter, chemical reactions give rise to Life by becoming subsumed within the organism's self-activity. Therefore, the divide between inorganic nature and organic nature is not an external boundary; as organic nature incorporates all ingredients of the chemical process, it confers chemical reactions with a new integration to constitute a biological entity [40] (p. 391). When the chemical process is complete, its stimulators cease to exist, so as not to spontaneously act once again. This is why self-renewal distinguishes life from the chemical process and this is why Hegel says that, if the products of the chemical process could spontaneously renew their activities, then they would be Life. To this extent, Life is a chemical process made perpetual [1] (p. 269). Thus, the chemical process is a middle term that unites inorganic and organic nature. Hegel speaks of this feature of the chemical process:

> The chemical process is the highest to which inorganic Nature can reach; in it she destroys herself and demonstrates her truth to be the infinite form alone. The chemical process is thus, through the dissolution of shape, the transition into the higher sphere of the organism where the infinite form makes itself, as infinite form, real... [1] (p. 271).

The establishment of this transition as a scientific fact is crucial to maintain the laws of dialectics as indispensable when grasping the motion in nature. The life emerging from the chemical processes signifies the transformation of one form of motion to another without any loss, and the dialectical law maintaining that the changes in quantity, at a specific nodal point, result in qualitative changes is affirmed. This is why, in a letter to Marx, Engels wrote that Hegel would be delighted with the discovery of the correlation of forces in physics [41] (p. 326). This law of motion, which was put forward by Hegel mainly in terms of speculative philosophy, now constitutes the basis of evolutionary biology that studies motion by way of dialectics and leads to further advances in this field of science. Pross mentions that an analysis of the materials that might have been formed in the prebiotic earth could offer some hints regarding the origin of life. He speaks of the experiments carried out by Stanley Miller, who formed a range of organic materials, including various amino acids, by mixing the four gaseous components thought at the time to be the main constituents of the prebiotic atmosphere and stimulating them by passing an electrical discharge through the mixture [38] (p. 93).

Pross's notion of life as a network is relevant to grasp the unity of inorganic and organic nature. This notion concerns two distinguishable aspects of life. The first is put forward by Pross in this way: "Life is just the resultant network of chemical reactions that emerges from the continuing cycle of replication, mutation, complexification, and selection, when it operates on particular chain-like molecules—in the case of life on Earth, the nucleic acids" [38] (p. 164). The second notion is that, although each human is composed of billions of individual cells, they consist of ten times as many bacterial cells as human ones. Therefore, from a numerical perspective, any human is more bacterial than human. This is why Pross calls human beings super-organisms—a giant network. Thus, every multicellular organism, rather than being a single living being, is an ecological network [38] (pp. 187–188).

These viruses present a compelling case when reflecting on the question of what life is. This reflection requires a discussion of the various postulations through which we define life. For instance, are complexification, the ability to realise metabolic processes, autonomous replication, and reproduction, sufficient premises to define an entity as a living being? The

viruses demonstrate a moment at which the distinction between organic and non-organic nature is blurred. According to Pross's definition of life, viruses should be regarded as non-living, as they lack the mechanisms to perform metabolic processes by themselves. A virus cannot reproduce and replicate by itself; it can realise these processes only through utilising the mechanisms of the host it has infected and, while doing this, its genetic code sometimes penetrates the code belonging to the host. Autonomous replication matters due to its close reciprocal relationships with other postulations regarding living beings, such as the being's capacity to gather energy, complexification, repetition, and the emergence of life as a network. Replication is a fundamental aspect of life as it leads to multiplication, which, according to Vladimir Vernadsky, is the essential feature distinguishing life from death [42] (p. 60).

The notion of the network brings forward the questions of organisation and complexity. Thompson maintains that the question of what life is should be researched in terms of the principles of biological organisation [21] (p. 92). Therefore, we happen to define life as the capacity of living beings to develop complex relationships with their exterior, and the complexification of the organs mediating these relationships. In fact, the notion of complexification may be taken to signify both the unity of the organic and the inorganic and to discern the former from the latter, since, as Vernadsky emphasises, "the structures of living organisms are analogous of those of inert matter, only more complex" [42] (p. 50).

Without a doubt, a virus is not an organism. However, it cannot be considered non-living because the notion of living must be understood not in terms of its independent and isolated being, but through its reciprocal relationships with its surroundings. This relatedness signifies viruses as beings epitomising the unity between the organic and the inorganic: they possess both the qualities of the living and non-living. They possess a semipermeable boundary. They can replicate and reproduce by utilising the relevant mechanisms of the host cell. When a virus infects the host, it becomes constitutive of the latter's body and changes the mode of operation of the cell that it has penetrated. Then, as Ted Grant and Alan Woods maintain, viruses "stand on the threshold of organic and non-living matter" [43] (p. 33). As a matter of fact, viruses are one of the bases of life. Ralph Buchsbaum [44] (p. 3) mentions the simple compounds of the elements that constitute the protoplasm, the living matter. Some of these gain the capacity to self-propagate, to realise additional combinations that are similar to them. This state of living matter resembles filterable viruses. They resume their activities even after repeated crystallisation. No one has succeeded in growing them in the absence of living matter, but viruses help to bridge the gap that was formerly thought to exist between non-living and living things; there is a gradual transition in complexity between them. Findlay calls viruses forms of quasi-life that connect the organic to non-organic [22] (p. 95).

Then, is it possible to consider nature a single organism? To reflect on this question Vernadsky's insights on the *notion of the biosphere* are crucial. His views and method of dealing with empirical evidence allow for an understanding of the biosphere as a single living being: it is, as such, the articulation of all life on earth. The wholeness of the organism is manifested in the notion of the biosphere: It is "a single orderly manifestation of the mechanism of the uppermost region of the planet — the Earth's crust" [42] (p. 39). The biosphere is "the terrestrial envelope that is occupied by the living matter, it is the entire field of existence of life" [42] (p. 118). Vernadsky proposes considering the empirical facts "from the point of view of a holistic mechanism that combines all parts of the planet in an indivisible whole" [42] (p. 39). According to him, living beings are parts of a harmonious cosmic mechanism [42] (p. 44). He considers the whole world to be a single living entity, and living beings as a single whole, as *epitomised* in the notion of *living matter*—the Earth's sum total of living organisms. Then, "all living matter can be regarded as a single entity in the mechanism of the biosphere" [42] (p. 58). After all, all living beings are genetically connected [42] (p. 88). Living matter continuously produces new chemical compounds, which extend the biosphere at an amazing speed. Therefore, the biosphere exists as a thick layer of new molecular systems [42] (p. 50). This entails the identification of the biosphere

with life, which manifests itself through chemical reactions: "There is no substantial chemical equilibrium on the crust in which the influence of life is not evident, and in which chemistry does not display life's work . . . Without life, the crustal mechanism of the Earth would not exist" [42] (p. 58). In this sense, one aspect of living beings, respiration, is determinative in the constitution of the biosphere. Vernadsky considers respiration to be an essential aspect of the mechanism of the biosphere. In this way, the existence of the ozone layer epitomises the unity between the organic and the inorganic. Vernadsky says that the formation of the ozone layer requires free oxygen, and the latter comes into existence solely through biochemical processes that would disappear if life were to stop [42] (p. 120).

*The notion of envelopment* is significant when comprehending the biosphere as a single whole entity and correlating the earth's crust with living matter. Vernadsky maintains that living matter clothes the whole terrestrial globe with a continuous envelope [42] (p. 60). Life encompasses the biosphere through adaptation, which has not yet reached its zenith. The domain of life encloses the biosphere: "There is no place in which it is unable to manifest itself one way or another" [42] (p. 117). An organism's range of existence and its adaptation abilities evince that the biosphere is a terrestrial envelope. For instance, the conditions that render life impossible simultaneously occur all over the planet [42] (p. 119). Therefore, one may conclude that the earth is animate.

## 6. Conclusions

Nature is present as becoming in the dialectical conception of Hegel. Fundamentally, the reciprocal relations of the beings that are active in Nature bring about this understanding of it. Only through its relationships within this whole, which is in constant motion, can the organism, the animal as a free subject, become intelligible. A fundamental aspect of this work is asserting that wholeness comes into view as the unity between the organism and its environment, both in the congruence of their coming into being and in their actual beings—as life in its concreteness.

Through the notion of nature, we can reach the true notion of freedom, since the former not only provides the conditions for freedom to be realised but also signifies freedom as motion, as Spirit, as the dismissal of the immediate bodily selfhood of an animal before it is reunited with the animal as physicality. Then, we reach the notion of the animal as infinite: It collects, assimilates, and unites the spurious infinity of its surroundings. This is what we call life, as proposed by Hegel: it is the freedom that is not only possible in and through this substance, but the freedom whose content is comprised of this substance. Therefore, throughout this work, I attempted to signify that the subjectivity of the animal is only possible through its interaction with the environment that it constructs.

Here, the fundamental issue is the objective impossibility of any organism evading its substance as nature. Both the extraversive character of the actions of the organism, which ensure its self-preservation, and the organism's emergence out of the ordinary chemical processes evince this impossibility. Then, to achieve a true understanding of this whole that is constant motion the intertwining natural sciences and philosophy is necessary. Hegel's philosophy of nature was grounded in the scientific discoveries of his time, but the scientific advances that have been made since then assert the unity between inorganic and organic nature with further rigour. Therefore, one can recognise that the founding of one science of both human and nature, through the process that Karl Marx spoke of [45] (p. 303) as natural science losing its abstractly material character and idealistic tendency to become the basis of human science, manifests itself as a necessary condition of scientific knowledge regarding the totality of life.

**Funding:** This research received no external funding.

**Institutional Review Board Statement:** Not applicable.

**Informed Consent Statement:** Not applicable.

**Data Availability Statement:** Not applicable.

**Conflicts of Interest:** The author declares no conflict of interest.

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
