# Peer review of "On the Understanding of the Unity of Organic and Inorganic Nature in Terms of Hegelian Dialectics"

_philosophies, doi:10.3390/philosophies7060128_

Round 1

Reviewer 1 Report

-The abstract should include some details as to what is original about this paper, what does it add to existing scholarship to orient the reader

-What does the author take an organism to be? Some either speculative or biological data should be provided (for example there are parasitic organisms, distributed organisms, what about a virus, is that an organism too?…) are they all included in the present definition of organism?

- Can nature as a whole (or biospheres) be considered an organism? How do these answers impact the basic definition on page 1, line 32-33

  • Why the shift to animal page 1, line 45. Are now animals suddenly identified with organisms?
  • References or quotes missing for claim in lines 171, 172
  • Line 261/262 grammar issues - rephrase
  • Especially part 3 (and to lesser extend part 4) is weaker than the other parts, the line of argument is less clear, the argumentation is less precise - it might be worth streamlining these two parts (or at least part 3), signposting conceptual moves better and further clarifying their role in the line of overall argument of the paper as a whole
  • Parts 5 and conclusion are again quite strong (but especially the input of part 3 (and by extension 4) is not clearly connected to part 5. This in my eyes would be the most interesting connection and would highlight the original contribution of this paper

Author Response

1) Now, a new sentence is included in the abstract to imply the originality of the work in terms of the unity of the sciences and the dialectics. 

2) Between lines 741-781 now a part on the question of whether viruses are living or not exists. I believe the discussion also provides some insights into what organic is.  

3) Lines 782-817 are on the notion of the biosphere. I attempted to discuss the biosphere as a living being through the work of Vernadsky.

4) It is mentioned in the text that the claim the reviewer mentioned (Line 172) is cited from the work of Michelini, Wunsch, and Stederoth.

5) I dealt with the grammar issue between lines 261-262.

These are the changes I was able to realise within the time I was given to complete the revision in line with the reviewer's comments.  

Reviewer 2 Report

The article is clear and the argumentation is solid. References to specialized literature are correct, although focused almost exclusively on works published in English.

The subject matter of the article is interesting and current. The understanding of the relationship between organic and inorganic nature in Hegel's philosophy presented by the author constitutes a contribution to Hegel studies.

For these reasons the article can be accepted in present form.

Author Response

I thank the reviewer for the comments. 

Reviewer 3 Report

I think the text is fine in its present form. I am not sure about the past tense in the Introduction.
Also, numbering of chapters seems unnecessary (or shouldn't include the introduction and conclusion).

Author Response

I fixed the problem of employing past tense throughout the introduction.